# Identification of *Bulinus forskalii* as a potential intermediate host of *Schistosoma hæmatobium* in Senegal

**Papa Mouhamadou Gaye**[1,2,3,4], **Souleymane Doucouré**[2], **Doudou Sow**[5], **Cheikh Sokhna**[2,3], **Stéphane Ranque** [2,3] *

**1** Aix-Marseille Université, IRD, AP-HM, SSA, VITROME, Marseille, France, **2** VITROME, Campus International IRD-UCAD de l'IRD, Dakar, Senegal, **3** Institut Hospitalo-Universitaire (IHU)-Méditerranée Infection de Marseille, Marseille, France, **4** Département Biologie Animale, Faculté des Sciences et Technique, UCAD de Dakar, Dakar, Senegal, **5** Department of Parasitology-Mycology, UFR Sciences de la Santé, Université Gaston Berger, Saint-Louis, Senegal

* stephane.ranque@univ-amu.fr

**Data Availability Statement:** All relevant data are within the manuscript and its Supporting Information files; all described nucleotide sequences have been deposited in the NCBI

## Abstract

Understanding the transmission of *Schistosoma hæmatobium* in the Senegal River Delta requires knowledge of the snails serving as intermediate hosts. Accurate identification of both the snails and the infecting *Schistosoma* species is therefore essential. Cercarial emission tests and multi-locus (COX1 and ITS) genetic analysis were performed on *Bulinus forskalii* snails to confirm their susceptibility to *S. hæmatobium* infection. A total of 55 *Bulinus forskalii*, adequately identified by MALDI-TOF mass spectrometry, were assessed. Cercarial shedding and RT-PCR assays detected 13 (23.6%) and 17 (31.0%), respectively, *Bulinus forskalii* snails parasitized by *S. hæmatobium* complex fluke. Nucleotide sequence analysis identified *S. hæmatobium* in 6 (11.0%) using COX1 and 3 (5.5%) using ITS2, and *S. bovis* in 3 (5.5%) using COX1 and 3 (5.5%) using ITS2. This result is the first report of infection of *Bulinus forskalii* by *S. hæmatobium* complex parasites in Senegal using innovative and more accurate identification methods to discriminate this snail and characterize its infection by S. hæmatobium.

## Author summary

Schistosomiasis is one of the most important neglected tropical diseases in the world. It is caused by blood flukes of the genus *Schistosoma*. In Senegal, the species *Schistosoma hæmatobium* is the most widespread and responsible for urinary Schistosomiasis in humans. Freshwater snails of the genus *Bulinus* including *Bulinus truncatus*, *Bulinus globosus*, *Bulinus senegalensis* and *Bulinus umbilicatus* are its main intermediate hosts. However, *Bulinus forskalii*, morphologically similar to *Bulinus senegalensis* is also frequently found but no study has ever mentioned its sensibility to *S. hæmatobium* in Senegal. The aim of our research is to reveal the relevance of *Bulinus forskalii* as a potential intermediate host of *S. hæmatobium* but also to provide an accurate identification of this snail. This study demonstrated the susceptibility of *Bulinus forskalii* to the *S. hæmatobium* parasite in

GenBank nucleotide data base. All reference nucleotide sequences used to identify snails and their parasites are from the NCBI GenBank nucleotide database and can be viewed by BLAST on http://blast.ncbi.nlm.nih.gov. The reference spectra of our in-house MALD-TOF MS database are available and can be downloaded with the following DOI number: https://doi.org/10.35088/f605-3922. The accession numbers detailed in the manuscript are: GenBank Accession No: OM535893.1, OQ860246, OQ860247 (MT579447.1, MT159594.1, MT580958.1, MT580959.1).

**Funding:** DS was funded by the Département Soutien et Formation, Institut de Recherche pour le Développement (https://www.ird.fr/), Grant Number: DA08022018. The funders had no role in study design, data collection and analysis, decision to publish, or preparation of the manuscript.

**Competing interests:** The authors have declared that no competing interests exist.

Senegal by using innovative and refined identification methods to discriminate this snail and to characterize its infection. This result significantly contributes to the improvement of our knowledge on host-parasite interactions and should be taken into account in future epidemiological studies and schistosomiasis control programs.

## Introduction

Snails of the genus *Bulinus* (Müller, 1781) are intermediate hosts for the larval development of trematodes parasite species of the *Schistosoma hæmatobium* species complex in Africa, the eastern Mediterranean, and Madagascar [1]. Several species of the S. *hæmatobium* group are endemic in Africa, notably *S. hæmatobium*, *S. intercalatum*, *S. guineensis*, *S. bovis* and *S. curassoni*. Only the first three are involved in human diseases [2,3]. Urogenital schistosomiasis, caused by *S. hæmatobium* (Bilharz, 1852), is present in most countries on the African continent [4], especially in all regions of Senegal [5]. *Schistosoma hæmatobium* is highly endemic in sub-Saharan Africa, where 600 million people are at risk of infection, and 200 million urinary schistosomiasis cases [6], including 150,000 deaths [7], are recorded *per* year.

In endemic areas, transmission of *S. hæmatobium* involves various freshwater gastropod snails species of the genus *Bulinus* [8–10]. In Senegal, *Bulinus truncatus*, *Bulinus senegalensis*, *Bulinus globosus* and *Bulinus umbilicatus* are the main intermediate hosts of *S. hæmatobium* [9,11,12]. *Bulinus forskalii* (Ehrenberg, 1831) occurs in the north in the Senegal River, in areas of permanent transmission of *S. hæmatobium* [10,13]. It is an area with perennial water (streams, ponds, irrigation networks…) and is the preferred habitat of *Bulinus forskalii* [14]. *Bulinus forskalii* is often misidentified as *Bulinus senegalensis*, a species with which it shares similar morphological features and often the same biotopes [15]. *Bulinus senegalensis* is found in abundance in areas with seasonal transmission, particularly in the center of the country [9], in cohabitation with *Bulinus forskalii*. The species *Bulinus senegalensis* shows a strong resistance to the drying up of temporary pools during drought compared to other intermediate host snails of Schistosomes [16].These two species are genetically similar and form, with *Bulinus camerunensis*, the *Bulinus forskalii* group [15,17]. The presence of *Bulinus forskalii* has been reported to be involved in *S. bovis*, *S. guineensis*, and *S. intercalatum* biological cycle in Western and Central Africa [3,18,19]. Whereas this *Bulinus* species has never been reported to be infected with *S. hæmatobium* in Senegal [20], previous studies have indicated that *Bulinus forskalii* could be a potential intermediate host for *S. hæmatobium* in Niger [21]. The main limitation of these finding was that the snails had been identified on morphological features. And identification errors cannot be excluded due to the known morphological similarity between *Bulinus forskalii* and *Bulinus senegalensis* [15].

The correct identification of intermediate host snails based on morphological and molecular (sequencing) criteria has limitations [22]. Morphological identification requires highy skilled experts, a specific documentation (good and updated identification keys) and also intact specimens. DNA-based identification is relatively time consuming costly, and the reference nucleotide sequence databases, notably Genbank, are not exhaustive.

Matrix-assisted laser desorption/ionization time-of-flight mass spectrometry (MALDI-TOF MS) has revolutionized microbial identification in clinical microbiology. It has also used been successfully used in entomology for rapid identification of many arthropods including mosquitoes, ticks, lice, fleas, and bed bugs [23–26]. More recently, this tool has been presented as an alternative tool for rapid identification of bivalve snails [27] and medically important gastropods [28]. Because MALDI-TOF MS is relatively fast, low-cost, it could be advantageously

used for the high-throughput rapid identification of snail populations in schistosomiasis control programs. Therefore, this study was aimed to assess the involvement of *Bulinus forskalii* as a potential intermediate host of *S. hæmatobium* and to provide an accurate identification of this snail.

## Methods

### Snail collection and morphological identification

Snails were collected as part of a malacological survey conducted in the Senegal River Delta (SRD) in September 2020. The SRD represents the terminal part of the Senegal River located in the northwest of the country (between latitudes 16˚/14˚40'N and longitudes 15˚30/16˚ 30'W), in the region of Saint-Louis. It is a marginal-littoral zone [29], covering an area of 6,000 km² (**Fig 1A**). Its density is 13.7 inhabitants/km² with a higher human concentration near the main water bodies [29,30]. Irrigated rice cultivation and artisanal fishing are the main activities carried out by the populations [16–18]. The SRD is characterized by a semi-desert climate with an average rainfall of approximately 300–400 mm [31]. The area has become endemic to schistosomiasis with perennial transmission dynamics [32] since the construction of the Diama and Manantali dams in the 1980s.

The snails were collected at water points located along the riverbanks. They were collected from aquatic plants, leaves, and dead branches in the water using a scoop net and a soft claw or by hand for some snails attach to the underside of aquatic vegetation. Snails from the same water point were placed together in the same pre-labelled containers (locality name, collection

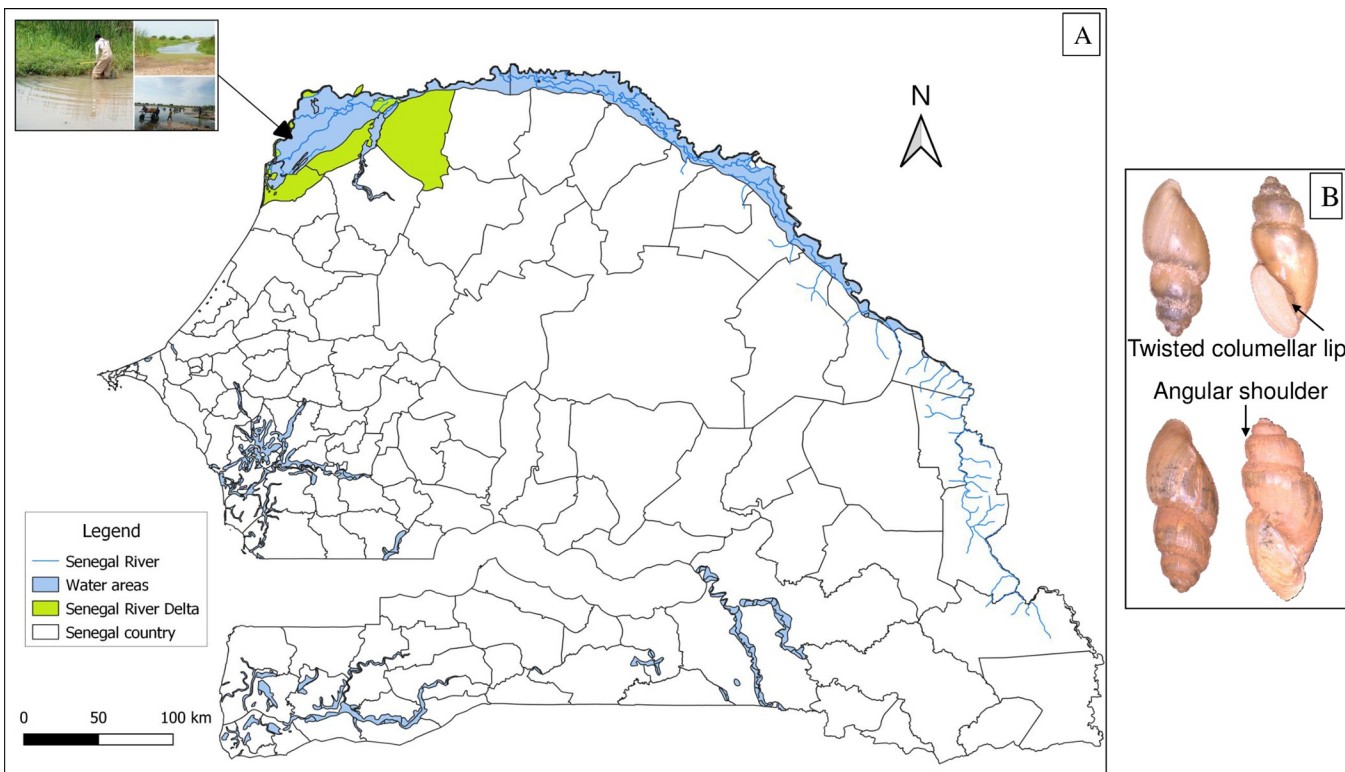

**Fig 1.** (**A**) A map of Senegal showing the snail sampling site, the Senegal River Delta (between latitudes 16˚/14˚40'N and longitudes 15˚30/16˚30'W), located in the Saint-Louis region. The map was produced using the Geographic Information System software QGIS v3.18.3-Zürich; http://www.qgis.org. (**B**) Morphological features of the *Bulinus forskalii* snail species analyzed in the current study (Sinister, brownish shell with a shoulder angle and twisted columellar lip).

date) containing cold water to inhibit cercarial excretion during transport. The taxonomic status of the collected snails was assessed by microscopy (Zeiss Axio Zoom.V16, Zeiss, Marly-le-Roi, France) using the Mandahl-Barth identification key [33] based on snail shell morphology. Several species were collected but only snails identified as *Bulinus forskalii* were included in this study. The species *Bulinus forskalii* presents a shell with a sinister opening, brownish to whitish, shouldered and strongly keeled with presence of a shoulder angle. The columellar lip is torse (**Fig 1B**).

### Cercarial shedding test

Cercarial shedding tests were then performed, by placing individual snails in the wells of a glass pillbox (24 wells) containing 5–10 ml of filtered water. All collected snails were then exposed to electric light for 30 min to 1 h to stimulate cercarial shedding. *Schistosoma* cercariae released from the snails were observed and identified morphologically using a binocular magnifying glass and a taxonomic key [34] but not kept (**Fig 2**). The snails were classified into infected and non-infected batches and stored in the laboratory at -80˚C.

### MALDI-TOF MS identification of snails

Each snail specimen (n = 55) was carefully extracted from its shell and dissected with a new sterile lame for Deoxyribo Nucleic Acid (DNA) extraction and Matrix Assisted Laser Desorption/Ionization-Time of Flight Mass Spectrometry (MALDI-TOF MS) analysis. The feet of the

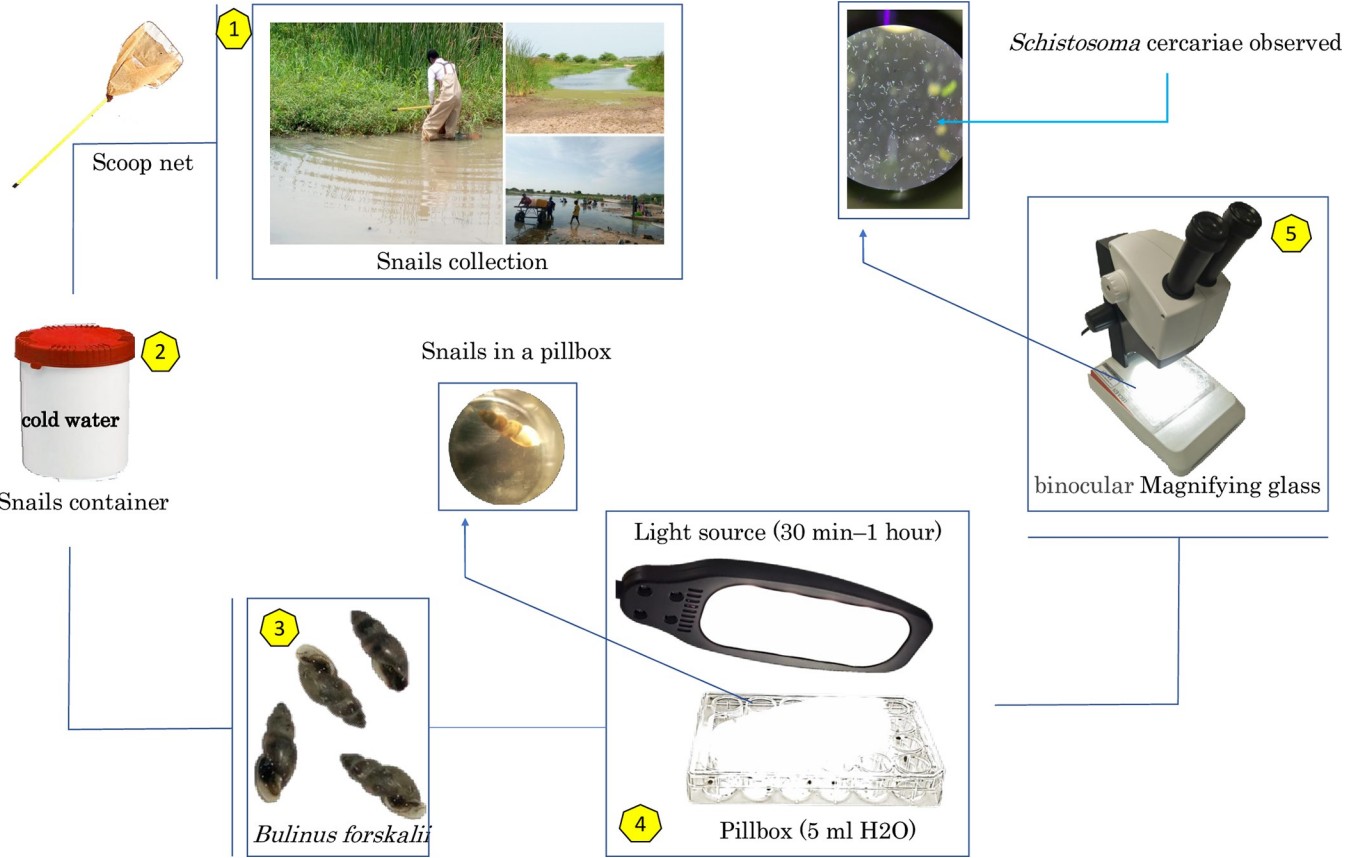

**Fig 2. An explanatory flow chart of the cercarial shedding protocol in intermediate host snails.**

snails were used for MALDI-TOF MS identification as described by Hamlili *et al*. (2021) [28]. The feet were successively rinsed with 70% ethanol and distilled water for 2 min and dried on sterile filter paper. They were then individually placed in 1.5 ml Eppendorf tubes with glass beads (Sigma, Lyon, France) and a mix containing 70% (v/v) formic acid (Sigma), 50% (v/v) acetonitrile (Fluka, Buchs, Switzerland), and high-quality liquid chromatography (HPLC) water. All samples were ground with 30µl of the mix using a TissueLyser II (Qiagen, Hilden, Germany) on three 1-min cycles at a frequency of 30 Hertz.

After centrifugation at 2000 rpm for 30 seconds, 1 µl of the supernatant from each homogenate was deposited on a MALDI-TOF MS target plate (Bruker Daltonics, Wissembourg, France) in 10 replicates. Each deposit was covered with one microlitre of a CHCA matrix suspension consisting of saturated α-cyano-4 hydroxycinnamic acid (Sigma, Lyon. France), 50% acetonitrile (v/v), 2.5% trifluoroacetic acid (v/v) (Aldrich, Dorset, UK), and HPLC grade water to allow co-crystallization. After drying for several minutes at room temperature, the target was introduced into the MALDI-TOF Microflex LT mass spectrometer (Bruker Daltonics, Breman, Germany) for analysis. To confirm the morphological identification of the snails, the MALDI TOF MS spectra obtained from the foot of each specimen were compared to the reference spectra from the in-house database (available at https://doi.org/10.35088/f605-3922) [28] using MALDI-Biotyper v3.0 software (Bruker Daltonics). The homemade reference spectra database already contains (n = 64) reference spectra of freshwater gastropod snails from Senegal (**S1 Table**). The blind test identification quality is estimated via log score values (LSV) (range 0 from 3), which quantifies the degree of identity between the query and the reference spectra in the database. A sample is considered correctly identified when the LSV value is $\geq$ 1.7.

A dendrogram was performed using MALDI-Biotyper v.3.0 software to visualize the level of heterogeneity of MS spectra from *Bulinus forskalii*, *Bulinus senegalensis* and *Biomphalaria pfeifferi* specimens as out-group. MS protein profiles from one to seven specimens of each snail species were randomly selected and used to create a dendrogram. Similarly Principal Component Analysis (PCA) performed using ClinProTools 2.2 software.

## DNA extraction

The remaining body of each specimen was used for genomic DNA extraction as a substrate to detect the parasite and confirm the molecular identity of the snails. DNA was extracted using the EZ1 DNA Tissue kit (Qiagen) following the manufacturer's recommendations. Each snail specimen was placed in a 1.5 mL Eppendorf tube and incubated at 56°C overnight in 180 µl of G2 lysis buffer (Qiagen Hilden, Germany) and 20 µl of proteinase K (Qiagen Hilden, Germany). The supernatant was recovered in another tube and then extracted using the EZ1 BioRobot extraction device (Qiagen Hilden, Germany). Genomic DNA from each sample was eluted with 200 µl of Tris-EDTA buffer (Qiagen) and stored at -20°C until use.

## Real-time PCR (RT-PCR) and PCR sequencing

Real-time PCR was used to detect *Schistosoma hæmatobium* group in all DNA extracts from *Bulinus forskalii* snails. The RT-PCR reaction targets the sequence of the highly repeated region Dra1 specific to the *S. hæmatobium* group [35]. The primers used Sh-FW and Sh-RV (**Table 1**), were identical to those initially described by Hamburger *et al*.[35] and the probe utilized was described by Cnops *et al*.[36]. The qPCR reaction was performed using a CFX96 thermal cycler (Bio-Rad, Marnes-la-Coquette, France) in a 20 µL reaction mixture containing 5 µL of DNA,10 µL of the Master Mix, 3.5 µL of sterile distilled water, and 0.5 µl of each of the primers and the TaqMan probe (Applied Biosystems, Foster City, CA, USA). Our

**Table 1. Primers used in the real-time PCR, *Schistosoma* parasite and snails sequencing protocols.** Primer sequences and names are given with the target marker, target organism and amplicon length.

| Primer names | Marker | Target organism | Length | Primer sequences (5'-3') | Authors |
|---|---|---|---|---|---|
| **Real-time PCR** | | | | | |
| Sh- FW | DraI | *Schistosoma hæmatobium* complex | 120 | GATCTCACCTATCAGACGAAAC | Hamburger *et al.* (2001) [35] |
| Sh- RV | | | | TCACAACGATACGACCAAC | |
| Probe (FAM 5'- 3'ZEN) | | | | TGTTGGTGGAAGTGCCTGTTTCGCAA | Cnops *et al.* (2013) [36] |
| **Schistosoma spp. sequencing** | | | | | |
| Asmit1 | COX1 | *Schistosoma* spp. | - | TTTTTTGGTCATCCTGAGGTGTAT | Webster *et al.* (2010) [37] |
| Sh.R | | *S. hæmatobium* | 543 | TGATAATCAATGACCCTGCAATAA | |
| Sb.R | | *S. bovis* | 306 | CACAGGATCAGACAAACGAGTACC | |
| ITS2_Schisto_F | ITS2 | *Schistosoma* spp. | ±369 | GGAAACCAATGTATGGGATTATTG | Schols *et al.* (2019) [38] |
| ITS2_Schisto_R | | | | ATTAAGCCACGACTCGAGCA | |
| **Snail sequencing** | | | | | |
| LCO1490 | COX1 | Snails | ±710 | GGTCAACAAATCATAAAGATATTGG | Folmer et al. (1994) [39] |
| HC02198 | | | | TAAACTTCAGGGTGACCAAAAAATCA | |

"±": possible variation in amplicon length due to interspecific variation

amplification program consisted of an initial 2-min denaturation step at 50˚C followed by a 3-min denaturation at 95˚C, then 40 cycles of 95˚C for 30 s and 60˚C for 1 min before holding the sample at 4˚C. For each qPCR plate, negative (sterile distilled water) and positive (*S. hæmatobium* egg DNA extract) controls were used. Samples with a cycle threshold (Ct) value less than 35 were considered positive.

Only RT-PCR positive samples were subjected to PCR sequencing of cytochrome C oxidase subunit I (COX1) [37] and the second internal transcribed spacer (ITS2) of the rRNA gene [38] to identify parasites within the *S. hæmatobium* group including *S. hæmatobium* and *S. bovis*. DNA obtained from each specimen was used for PCR reaction. The amplification reaction was performed in a thermal cycler (Applied Biosystems, 2720, Foster City, U.S.A.) with AmpliTaq Gold 360 PCR master mix (Applied Biosystems, Waltham, MA., U.S.A.) using universal forward primer Asmit1 and reverse primers (Sh and Sb) and ITS2 with primers Schisto_F and Schisto_R (**Table 1**) [37,38]. The amplification protocol was carried out with initial denaturation at 95˚C for 15 min, with 39 cycles at 95˚C for 30 s, 58˚C for 1 min (at 56˚C for 1 min 30 s for ITS2), 72˚C for 1 min, and a final step at 72˚C for 7 min.

The amplified products were purified using a Macherey Nagel plate (NucleoFast 96 PCR, Düren, Germany) and sequenced using the same primers. Sequencing was performed using BigDye Terminator v1.1, v3.15x Sequencing Buffer (Applied Biosystems, Warrington, UK) and run on an ABI 3100 automated sequencer (Applied Biosystems). The resulting sequences were assembled and corrected using Chromas Pro v.1.77 (Technelysium Pty. Ltd, Tewantin, Australia) and BioEdit v. 7.0.5.3 to be used for a Basic Local Alignment Search Tool (BLAST) search on the National Center for Biotechnology Information (NCBI) online database (http://blast.ncbi.nlm.nih.gov).

The *Bulinus forskalii* MS identification was confirmed by nucleotide sequence-based analysis of the Folmer region of COX1. The same protocol as the one above was applied for sequencing the Folmer region using universal primers LCO1490 and HCO2198 (**Table 1**) [39]. However, the PCR amplification program here consists of an initial denaturation at 95˚C for 15 min, with 40 cycles at 95˚C for 30s, at 40˚C for 30 s, 72˚C for 1min 30s and a final stage at 72˚C for 7 min.

## Results

### MS identification of snails

All snails were identified as *Bulinus forskalii* by MALDI-TOF MS with log scores ranging from 1.89 to 2.63 [mean (SD) = 2.31 (0.20)] and reproducible spectra, indicative of a reliable identification (**S1 Fig**). Dendrogram analysis revealed specific clustering on distinct branches of snails by species. The specimens of the same species were clustered in the same part of the MSP dendrogram (**Fig 3A**). Principal component analysis (PCA) using ClinProTools 2.2 software showed that snail specimens of the same species cluster together (**Fig 3B**).

### Molecular identification of snails and detection of *S. hæmatobium* complex infection

*Bulinus forskalii* identification were further confirmed by 100.00% identity with the COX-Folmer of *Bulinus forskalii* (GenBank Accession No: OM535893.1) [3,22]. In the 55 *Bulinus forskalii* specimens, 13 (23.6%) and 17 (30.9%) were infected with *Schistosoma* parasites with the cercarial shedding tests or the Dra1-RT-PCR, respectively (**S2 Table**). The percentage of infected snails observed by the cercarial shedding test did not significantly differ from that obtained with real-time PCR ($\chi2 = 0.41$, df = 1, p = 0.52). The RT-PCR was positive (Cycle threshold (Ct) $\leq$ 35) in each snail that shed cercariae (**S2 Table**).

### DNA sequence-based identification of *Schistosoma* spp. parasites

The COX1 nucleotide sequence further identified six (10.9%) *S. hæmatobium*, and three (5.5%) *S. bovis*. In contrast, ITS2 nucleotide sequence based identification yielded three (5.5%) for both *S. hæmatobium* and *S. bovis*. BLAST query of the NCBI nucleotide database identified either *S. hæmatobium* or *S. bovis*, as detailed in **Table 2**.

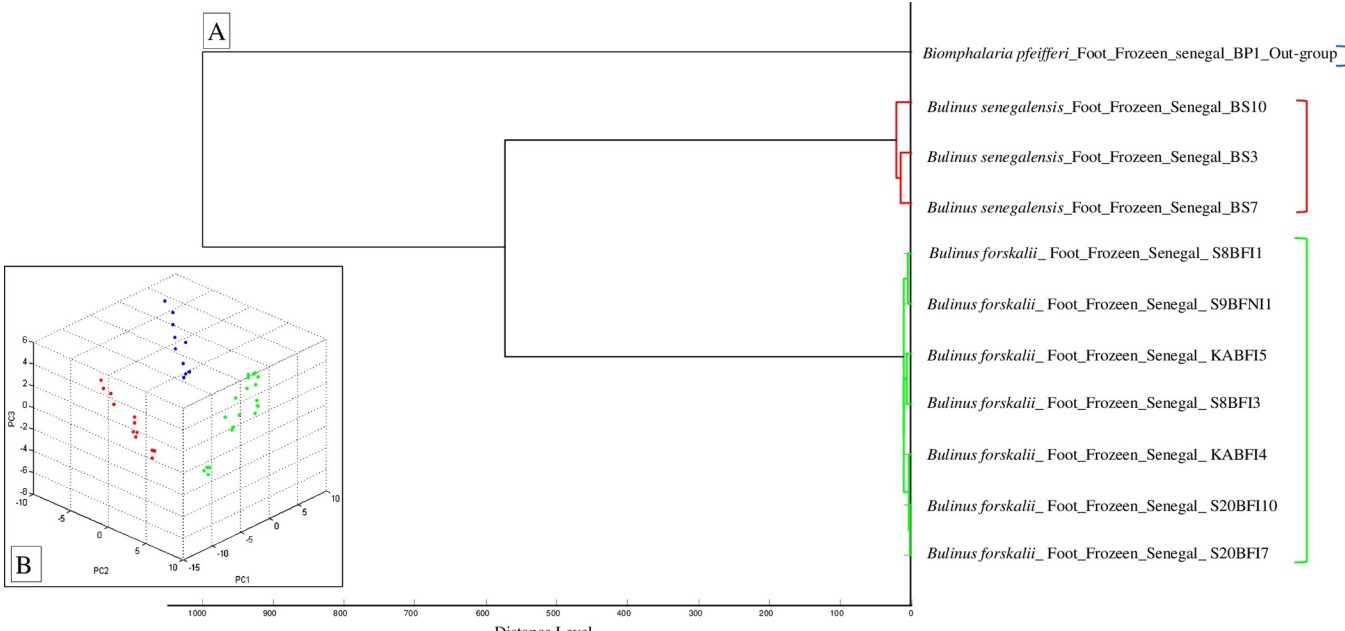

**Fig 3.** (A) Dendrogram constructed using MALDI-Biotyper v.3.3 software including representative random MS spectra of *Bulinus forskalii* (green), *Bulinus senegalensis* (red) and *Biomphalaria pfeifferi* (blue). (B) Principal component analysis (PCA) performed on spectra of *Bulinus forskalii* (green), *Bulinus senegalensis* (red) and *Biomphalaria pfeifferi* (blue) using ClinProTools 2.2.

**Table 2. MALDI-TOF MS identification of the snail parasitized by *S. hæmatobium* complex fluke, and COX1 and ITS2 DNA sequence-based identification of the *Schistosoma* spp. parasites.**

| Samples ID | Snail species ID by MS | LSV | cercarial shedding test* | RT-PCR (Ct) | Schistosomes species ID by Cox1 (% Identity) | GB Accession No** | Schistosomes species ID by ITS2 (% identity) | GB Accession No** |
|---|---|---|---|---|---|---|---|---|
| KABfI4 | *Bulinus forskalii* | 2.37 | (+) | 29.43 | *S. hæmatobium* (100.00%) | MT579447.1 | NA*** | - |
| KABfI5 | *Bulinus forskalii* | 2.60 | (+) | 29.23 | *S. hæmatobium* (99.78%) | MT579447.1 | NA | - |
| KABfNI5 | *Bulinus forskalii* | 1.89 | (-) | 28.57 | *S. hæmatobium* (99.78%) | MT579447.1 | NA | - |
| S20BfI7 | *Bulinus forskalii* | 2.55 | (+) | 23.01 | *S. hæmatobium* (99.78%) | MT579447.1 | *S. bovis* (100.00%) | MT580958.1 |
| S20BfI9 | *Bulinus forskalii* | 2.56 | (+) | 31.17 | *S. hæmatobium* (99.78%) | MT579447.1 | NA | - |
| S20BfI10 | *Bulinus forskalii* | 2.63 | (+) | 25.00 | NA | - | *S. hæmatobium* (100.00%) | MT580959.1 |
| KSD2BfI1 | *Bulinus forskalii* | 2.29 | (+) | 19.39 | *S. bovis* (99.23%) | MT159594.1 | *S. bovis* (100.00%) | MT580958.1 |
| KSD2BfI4 | *Bulinus forskalii* | 2.38 | (+) | 25.41 | *S. bovis* (99.61%) | MT159594.1 | *S. bovis* (100.00%) | MT580958.1 |
| S9BfNI1 | *Bulinus forskalii* | 2.31 | (-) | 34.94 | *S. hæmatobium* (100.00%) | MT579447.1 | *S. hæmatobium* (100.00%) | MT580959.1 |
| SOBfNI1 | *Bulinus forskalii* | 2.11 | (-) | 33.88 | *S. bovis* (99.61%) | MT159594.1 | *S. hæmatobium* (100.00%) | MT580959.1 |

*Cercarial shedding test: (+), positive; (-), negative.

** Accession numbers of the Nucleotide GenBank sequences with the highest identity percentage

*** NA: not available, because of sequencing failure

## Discussion and conclusion

Studies aiming to detect and identify parasites of the *S. hæmatobium* group in snails are scarce in Senegal. Our findings demonstrate that *Bulinus forskalii* can be parasitized by *S. hæmatobium* complex fluke, in particular, *S. hæmatobium* s.s., *S. bovis* s.s and by *S. hæmatobium* x *S. bovis* hybrids. Our findings contrast with those of Tian-Bi *et al*. [40] who found no infected *Bulinus forskalii* based on cercarial shedding tests. Nevertheless, studies conducted in *Bulinus senegalensis* and *Bulinus umbilicatus* in central Senegal have shown similar results [11]. The evidence of *S. hæmatobium* cercariae excretion by *Bulinus forskalii* demonstrates that this snail is an intermediate host, in which the parasite develops from miracidium to the cercaria.

In our study, all the snails that were positive in the shedding test were also positive in the RT-PCR, but the reverse was not true. This could be explained by the fact that parasites are excreted only at maturity in the shedding test whereas RT-PCR is able to detect them at all stages of their development in the snail. Our results do not show a statistically significant difference between the two techniques, which could be related to our relatively small sample size. However, other studies have shown that the cercarial shedding test largely underestimates snail infection when compared to PCR-based assays [41–43].

Identification of snail hosts is essential for mapping the transmission and risk of transmission of human and animal schistosomiasis. Some snail specimens, especially those belonging to related species such as *Bulinus forskalii* and *Bulinus senegalensis*, are often difficult to identify as mentioned in some previous studies [14,15]. Here, *Bulinus forskalii* was identified by sequencing and MALDI-TOF MS. The latter is a robust identification tool validated in several snail species [28] which might be particularly advantageous in large epidemiological surveys.

Morphological misidentification may also be related to invisible characters of snail specimens or to changes in their shells induced by environmental parameters. In our study, all intact and good quality specimens of *Bulinus forskalii* with well-preserved shells were correctly identified by morphology but had to be confirmed by another identification method, MALDI-TOF MS. The use of MALDI-TOF MS thus removed any ambiguity related to the identification of our snails with the advantage of economizing on time and consumables. Previous studies of the same type have presented ambiguities or errors in the identification of the species *Bulinus forskalii*. A study by McCullough *et al.* [44] identified snails involved in the transmission of *S. hæmatobium* and *S. bovis* as *Bulinus forskalii*, however, their species status was later confirmed as *Bulinus senegalensis* by Smithers [45]. This shows the importance of rapid and more reliable identification tools including MALDI-TOF MS.

In numerous previous studies, the snail species *Bulinus forskalii* had not been identified as a natural *S. hæmatobium* host in Senegal [12,14,32]. The evidence that *Bulinus forskalii* can be an intermediate host of *S. hæmatobium* complex parasites which are in the main schistosomiasis agents in Senegal, is a significant advancement of knowledge on host-parasite interactions. It should be taken into account in further epidemiological surveys and schistosomiasis control programs. The main limitation of our study is that discrepant ITS2 and COX1-based identification in a given snail specimen can be interpreted either as an infection with one hybrid parasite or a co-infection with two distinct parasite species. Further studies on isolated cercariae should give a final answer this open question.

## Supporting information

**S1 Table. List of Gastropod species present in our homemade MALDI-TOF MS database.**
(DOCX)

**S2 Table. Detailed results of MALDI-TOF MS identification, cercarial emission test and RT-PCR detection of *S. hæmatobium* complex performed of all snails.**
(DOCX)

**S1 Fig.** (A) Distinction of MALDI-TOF MS spectra of *Bulinus forskalii* (blue) and *Bulinus senegalensis* (red), illustrated by the spectral profiles of some specimens. a.u.: arbitrary units; m/z: mass/charge ratio. (B) Superimposed spectra of *Bulinus forskalii* (blue) and *Bulinus senegalensis* (red) with visually observed discriminating peaks indicated by black arrows.
(TIF)

## Acknowledgments

The authors would like to thank the population of the Senegal River Delta for their support.

## Author Contributions

**Conceptualization:** Papa Mouhamadou Gaye, Cheikh Sokhna, Stéphane Ranque.

**Data curation:** Papa Mouhamadou Gaye.

**Funding acquisition:** Souleymane Doucouré, Doudou Sow, Cheikh Sokhna, Stéphane Ranque.

**Investigation:** Papa Mouhamadou Gaye.

**Methodology:** Papa Mouhamadou Gaye, Cheikh Sokhna, Stéphane Ranque.

**Project administration:** Cheikh Sokhna, Stéphane Ranque.

**Resources:** Cheikh Sokhna, Stéphane Ranque.

**Supervision:** Cheikh Sokhna, Stéphane Ranque.

**Validation:** Souleymane Doucouré, Doudou Sow, Cheikh Sokhna, Stéphane Ranque.

**Visualization:** Souleymane Doucouré, Doudou Sow, Cheikh Sokhna.

**Writing – original draft:** Papa Mouhamadou Gaye.

**Writing – review & editing:** Souleymane Doucouré, Doudou Sow, Cheikh Sokhna, Stéphane Ranque.

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
