## [Decision Letter · Decision Letter 0]

16 Aug 2022

Dear Stephane Ranque,

Thank you very much for submitting your manuscript "Identification of Bulinus forskalii as a potential intermediate host of Schistosoma hæmatobium in Senegal" for consideration at PLOS Neglected Tropical Diseases. As with all papers reviewed by the journal, your manuscript was reviewed by members of the editorial board and by independent and expertise reviewers. In light of the reviews (below this email), we would like to invite the resubmission of a significantly-revised version that takes into account the reviewers' comments. 

Your manuscript has been assessed by 3 expertise reviewers in schistosomiasis research. The manuscript reports the novel information of Bulinus forskalii snail and its potential to be S. haematobium intermediate host in the endemic area. Even through, there is no additional experiments suggested from reviewers, but there are major concerns of the sample size and insufficient statistic used in the current version. All the reviewers suggested to improve the statistic used in the study along with other questions/comments to clarify the materials and methods, and result sections. Additional information in the introduction especially MALDI-TOF MS including the related-references will improve the manuscript as well. 

Please clarify the reviewer comments, suggestions and/or questions, then and address them point-by-point that would improve the manuscript.

We cannot make any decision about publication until we have seen the revised manuscript and your response to the reviewers' comments. Your revised manuscript is also likely to be sent to reviewers for further evaluation.

Sincerely,

Wannaporn Ittiprasert, Ph.D

Academic Editor

Makedonka Mitreva

Section Editor

PNTD-D-22-00788

Identification of Bulinus forskalii as a potential intermediate host of Schistosoma haematobium in Senegal

Dear Dr. Stephane Ranque,

Thank you for submitting your manuscript ‘Identification of Bulinus forskalii as a potential intermediate host of Schistosoma haematobium in Senegal’ to PLoS NTD. Your manuscript has been assessed by 3 expertise reviewers in schistosomiasis research. The manuscript reports the novel information of Bulinus forskalii snail and its potential to be S. haematobium intermediate host in the endemic area. Even through, there is no additional experiments suggested from reviewers, but there are major concerns of the sample size and insufficient statistic used in the current version. All the reviewers suggested to improve the statistic used in the study along with other questions/comments to clarify the materials and methods, and result sections. Additional information in the introduction especially MALDI-TOF MS including the related-references will improve the manuscript as well. 

Please clarify the reviewer comments, suggestions and/or questions, then and address them point-by-point that would improve the manuscript.

Best regards,

Wannaporn Ittiprasert, Ph.D

PLoS NTD

Reviewer's Responses to Questions

**Key Review Criteria Required for Acceptance?**

**Methods**

-Are the objectives of the study clearly articulated with a clear testable hypothesis stated?

-Is the study design appropriate to address the stated objectives?

-Is the population clearly described and appropriate for the hypothesis being tested?

-Is the sample size sufficient to ensure adequate power to address the hypothesis being tested?

-Were correct statistical analysis used to support conclusions?

-Are there concerns about ethical or regulatory requirements being met?

Reviewer #1: (No Response)

Reviewer #2: -Are the objectives of the study clearly articulated with a clear testable hypothesis stated?

The objectives and hypothesis have been implicitly stated. These could be explicitly stated in an additional paragraph in the introduction.

-Is the study design appropriate to address the stated objectives?

Yes.

-Is the population clearly described and appropriate for the hypothesis being tested?

While the study focuses on B. forskalii, inclusion and/or discussion of other snail species found in the field sampling area would strengthen the manuscript.

-Is the sample size sufficient to ensure adequate power to address the hypothesis being tested?

Yes, assuming the hypothesis is something to the effect of "B. forskalii snails that reside in the delta region of the Senegal River can carry S. haematobium infections". The current study conducted experiments on a smaller scale in terms of geographical location compared to other referenced studies of intermediate hosts of Schistosoma parasites.

-Were correct statistical analysis used to support conclusions?

No statistical analyses were described. Perhaps there could be a test to correlate cercarial shedding to Ct value or COX1 and ITS2 (yes/no) status.

-Are there concerns about ethical or regulatory requirements being met?

No, there are no ethical or regulatory concerns.

-Other comments

Consider adding information to the methods section to answer the following questions: How was the cercarial shedding test done? How was the DNA extracted? How was the RT-PCR conducted? How was the PCR sequencing conducted?

Reviewer #3: P.M. Gaye et al build on their previous work from 2021 where they use MALDI-TOF MS to identify host snails. Here they collect 55 Bu forskalli from the Senegal river delta and look for evidence of S.haemotobium infection. The methods section of the paper is not sufficient to describe their study, but can be easily improved upon. 

1) Please describe in more detail the area of the delta where the Bu forskalli was collected; a map may be helpful here.

2) Include in your methods how the snails were held (under what conditions) until the experiment to determine the presence of Sh or other parasites.

3) Describe your shedding protocol. How were the cercariae induced from the Bu forskalli?

4) On line 75, the authors tell us they used a reference spectra database and provide the link. Please describe this database as it was not intuitive and this Reviewer could not find the relevant page for snail references.

**Results**

-Does the analysis presented match the analysis plan?

-Are the results clearly and completely presented?

-Are the figures (Tables, Images) of sufficient quality for clarity?

Reviewer #1: Improve figure 1 quality.

Reviewer #2: -Does the analysis presented match the analysis plan?

Yes.

-Are the results clearly and completely presented?

As suggested below, more of the data could be made available, such as the results for all 55 B. forskalii specimens in Table 1.

-Are the figures (Tables, Images) of sufficient quality for clarity?

Yes. See below for suggestions.

Figure 2: If possible, consider showing (or explicitly labeling) the MALDI-TOF MS spectra of the reference sample for B. forskalii and, if available, those of other Bulinus members. Also consider showing the corresponding log score values for each spectra.

Table 1: The text describes 55 samples, while this table shows 10. How were these chosen? Are these specimens that met the Ct cutoff of 35 for the RT-PCR? Consider clarifying this in the text and/or caption. These results also do not show completely uninfected specimens, that is, specimens with a negative shedding test, CT>35, and negative sequencing results for COX1 and ITS2. Perhaps there were some, but they are not shown in the table? Consider including the data for all tested specimens, if not in this table, as supplementary information.

-Other comments

On line 67, the manuscript states that "[the authors] collected (n=55) snails from the Senegal River Delta area, identified as B. forskalii". Were there other Bulinus snail species present but not considered for analysis? While perhaps outside the scope of the study, such information might help describe the contribution of B. forskalii relative to other Bulinus species as an intermediate host for S. haematobium.

Reviewer #3: The authors claim in their results that the Bu forskalli were identified using MALDI-TOF MS, but also confirmed by COX-Folmer. Can they include that data in this manuscript?

**Conclusions**

-Are the conclusions supported by the data presented?

-Are the limitations of analysis clearly described?

-Do the authors discuss how these data can be helpful to advance our understanding of the topic under study?

-Is public health relevance addressed?

Reviewer #1: (No Response)

Reviewer #2: -Are the conclusions supported by the data presented?

Yes.

-Are the limitations of analysis clearly described?

Yes.

-Do the authors discuss how these data can be helpful to advance our understanding of the topic under study?

There could be more discussion about how this new knowledge that B. forskalii can carry S. haematobium infection can help improve epidemiological studies, surveillance efforts, and contribute to controlling schistosomiasis. For example, in surveillance studies, have researchers previously dismissed the possibility that B. forskalii snails could carry S. haematobium, and thus would assess a region to be at low risk of S. haematobium infections if only B. forskalii were found? If so, the findings in this study would encourage researchers to re-evaluate the guidelines for this hypothetical assessment.

-Is public health relevance addressed?

Yes, but, as suggested above, it could be more thoroughly discussed.

-Other comments

On line 112, the text asserts "Our findings demonstrate that B. forskalii can be parasitised by S. hæmatobium complex fluke, in particular, S. hæmatobium s.s., S. bovis s.s., and hybrids between them." The results do not necessarily demonstrate that S. haematobium x S. bovis hybrid species were detected in these snails. As stated on line 124, "discrepant ITS and COX-based identification in a given snail specimen can be interpreted either as an infection with one hybrid parasite or a co-infection with two distinct parasite species." Consider slightly modifying the statement to the effect of "Our findings demonstrate that B. forskalii can be parasitised by S. hæmatobium complex fluke, in particular, S. hæmatobium s.s., S. bovis s.s., and possibly S. haematobium x S. bovis hybrids."

Reviewer #3: The authors' conclusions are supported by the data presented.

**Editorial and Data Presentation Modifications?**

Reviewer #1: Introduction:

There is some data of B. forskalii and B. senegalensis distribution in Senegal? If so consider adding this information as it is extremely relevant to this study as some of B. senegalensis could be misidentified due to the similarity of the species.

I suggest you add some information about the MALDI-TOF MS as a research tool and its relevance to several species’ identification.

Maybe you can add a sentence with the relevance of the present study at the end of the introduction.

Methods:

Since according to Hamlili FZ et. al, 2021 it is necessary a specific database depending on the storage condition and can impact the quality of the tool I suggest you indicate the storage condition performed in the present study. You can indicate that the methods were performed as Hamlili FZ et. al, 2021, but quickly describe it.

The same I can point to the other methods - genomic DNA extraction, nucleotide sequence-based analysis, Real-time-Polymerase Chain Reaction (RT-PCR), and PCR sequencing - readers need to know exactly the methods you performed (reproducible study).

Results:

What is the relevance of the MALDI-TOF MS technique if the identification differentiation between B. forskalii and B. senegalensis can be performed by molecular identification using the COI gene?

To improve the results view and comparison I think that characteristic MALDI-TOF MS spectra acquired from B. senegalensis feet should be added to the Figure 1 as a control.

Can the infected snails be different from the non-infected under the MALDI-TOF MS analysis?

Reviewer #2: Line 70: "Desoxyribo" should be Deoxyribo

Line 72: ".. of Flight (MALDI-TOF MS)": Consider ".. of Flight Mass Spectrometry (MALDI-TOF MS)"

Line 31: "Nucleotide sequence analysis identified 6 (11.0%), using COX1, and 3 (5.5%), using ITS2, S. hæmatobium, and 3 (5.5%) S. bovis" could be clarified to something to the effect of "Nucleotide sequence analysis identified 6 (11.0%) using COX1 and 3 (5.5%) using ITS2 for S. hæmatobium, and 3 (5.5%) using COX1 and 3 (5.5%) using ITS2 for S. bovis"

On line 32, the manuscript asserts that "This result is the first report of infection of B. forskalii by S. hæmatobium complex parasites." However, the study by Labbo and coworkers (reference 19) have previously reported B. forskalii with S. haematobium infection. The statement could be modified by appending "in Senegal" or otherwise rewritten to reflect that the current study, in contrast to the previous study, uses more precise methods to differentiate between B. forskalii and B. senegalensis in identifying B. forskalii infected with S. haematobium.

Reviewer #3: The paper is well written; no editorial suggestions.

**Summary and General Comments**

Reviewer #1: The present study is relevant since it is the first report of B. forskalii infected by S. hæmatobium complex parasites in Senegal. Although it improves the knowledge about host-parasite interactions it also highlights that more similar studies should be held in endemic sites to better understand the transmission of Schistosoma species. It is relevant that MALDI-TOF mass spectrometry analysis was applied again as a tool that allows the differential identification of B. forskalii and B. senegalensis snails since B. senegalensis is already one of the main intermediate hosts of S. hæmatobium.

Reviewer #2: -Summary of study

The manuscript describes a study of 55 B. forskalii snails with respect to S. haematobium and S. bovis infection in the delta region of the Senegal River in September 2020. The study addresses the challenge of differentiating B. forskalii and B. senegalensis by using MALDI-TOF MS and PCR sequencing (COX-Folmer primers) analyses and follows with a cercarial shedding test and PCR sequencing analyses for S. haematobium COX1 and ITS2, showing that B. forskalii of this region can carry S. haematobium, S. bovis, or possibly S. haematobium x S. bovis hybrid parasites.

-Strengths

The analysis methods (MALDI-TOF MS and PCR sequencing) increase the confidence that snail specimens represent B. forskalii, not B. senegalensis, and that these B. forskalii snails are carrying S. haematobium parasites.

-Weaknesses

Field sampling included a single region in Senegal during a limited temporal window in September 2020. Sampling of more regions over a larger temporal period would increase the confidence that B. forskalii, under a broader range of conditions and environments, can carry S. haematobium infection.

Reviewer #3: The authors' finding is important for the field of Schistosomiasis, particularly in endemic areas because they show definitively that Bu forskalli can host Sh. This finding will be of interest to parasitologists and malacologists; not only for the technology used to draw the study's conclusions, but also because of the implications for human infection. My "major revision" recommendation is mainly to address deficiencies in the Methods section, but does not necessarily require new experiments.

PLOS authors have the option to publish the peer review history of their article (what does this mean?). If published, this will include your full peer review and any attached files.

Reviewer #1: No

Reviewer #2: Yes: Kenji Ishida

Reviewer #3: No
---

## [Decision Letter · Decision Letter 1]

27 Feb 2023

Dear Stephane Ranque,

Thank you for submitting your revised manuscript. The revised version has been backed to assess by expertise reviewers. The revised manuscript have great improvement to clarify the information in materials and methods, statistic used and others. However, there is still minor comments/suggestion to clarify the number of specimens used, especially in table 3 and discuss in depth for the advantage of MADOI-TOF-MS method rather than morphology identification.

Sincerely,

Wannaporn Ittiprasert, Ph.D

Academic Editor

Makedonka Mitreva

Section Editor

Reviewer's Responses to Questions

**Key Review Criteria Required for Acceptance?**

**Methods**

-Are the objectives of the study clearly articulated with a clear testable hypothesis stated?

-Is the study design appropriate to address the stated objectives?

-Is the population clearly described and appropriate for the hypothesis being tested?

-Is the sample size sufficient to ensure adequate power to address the hypothesis being tested?

-Were correct statistical analysis used to support conclusions?

-Are there concerns about ethical or regulatory requirements being met?

Reviewer #1: Sentence lines 160-162 is duplicated on lines 163-166. (manuscript with track changes)

Table 1 consider change it to Supplementary data.

Reviewer #2: (No Response)

Reviewer #4: The objetives are clearly articulated with a clear testable hypothesis stated

Sample size is somewhat small. This limits the statistical analysis.

Reviewer #5: 1- Line 100: Fig1 could not be downloaded in high resolution

2- Line 115: Please add the genomic coordinates of the area in which the snails were sampled

3- Line 119: Please make it clear which morphological features of the Bulinus forskalii snail species were analyzed 

4- Line 145: sentence is duplicated, needs correction

5- Table 1 needs correction: some terms are truncated (Ethanol absolu) or incorrect (Coordonates), etc

6- Line 180: please specify what has been used as negative and positive controls

**Results**

-Does the analysis presented match the analysis plan?

-Are the results clearly and completely presented?

-Are the figures (Tables, Images) of sufficient quality for clarity?

Reviewer #1: Line 264 change paratised to parasitized. (manuscript with track changes)

Why on table 3 do you only have 10 specimens identified on the table? On lines 259 - 260 you indicate that 13 were positives through shedding and 17 through RT-PCR.

Reviewer #2: (No Response)

Reviewer #4: (No Response)

Reviewer #5: The quality of the figures should be improved.

**Conclusions**

-Are the conclusions supported by the data presented?

-Are the limitations of analysis clearly described?

-Do the authors discuss how these data can be helpful to advance our understanding of the topic under study?

-Is public health relevance addressed?

Reviewer #1: Line 283 change paratised to parasitized. (manuscript with track changes)

Reviewer #2: (No Response)

Reviewer #4: (No Response)

Reviewer #5: Line 250: Has the authors tried/identified sporocystes in the collected Bulinus forskalii snails?

**Editorial and Data Presentation Modifications?**

Reviewer #1: (No Response)

Reviewer #2: (No Response)

Reviewer #4: (No Response)

Reviewer #5: (No Response)

**Summary and General Comments**

Reviewer #1: I suggested minor revision just to adjust some details and regarding the Table 3 that I still didn’t understand why only consider 10 specimens to add on the table 3.

Reviewer #2: The authors have adequately addressed the reviewers' comments and have improved the quality of the manuscript in this revision. The manuscript is suitable for publication.

Reviewer #4: The paper presented by Mouhamadou et al, entitled Identification of Bulinus forskalii as a potential intermediate host of

 Schistosoma haematobium in Senegal, describes a field study for the collection and identification of B. foskalii snails for infection with S. haematobium and S. bovis in an area of the Senegal River during the month of September 2020. 

The finding that B. forskalli snails may be intermediate hosts of S. haematobiun is of great interest for the control of schistosomiasis in Sengegal.

The manuscript is well written and structured and allows a good understanding. 

Methodologically it is correct, but I would like the authors to discuss in more depth the advantages of the MADI-TOF MS technique for the identification of snails, taking into account that they have previously been identified without problems by morphological characters. They comment in the introduction that morphological identification requires a great deal of expertise, that DNA identification is time-consuming and that there is not much information in databases. In schistosomiasis endemic areas, is it feasible to make a MALDITOF ? It also requires knowledge and experience to do it, as well as the databases to make a good comparison of identifications. Please try to explain this in more detail. Maybe there are not so many advantages compared to an identification of snails by morphology.

Typo in Line 188; S. bovis should be written in italics.

Reviewer #5: The authors provide a report that is centered around the identification/description of Bulinus forskalii snails that can host S. hæmatobium in a limited region of the Senegal River. The manuscript has been greatly improved after the first revision: the descriptions are clearer, the results better presented now, and additional discussions improved the interpretations. I have two main concerns:

1- the fact that a single, limited region in Senegal was sampled in a short temporal window (September 2020). Additional sampling in other regions over a larger period could strengthen the authors’ claim. 

2- a demonstration that Bulinus forskalii can be infected by S. hæmatobium outside the field, in the laboratory could strengthen the manuscript. Has this been already performed, or is it possible?

Other points that should be considered are below:

Introduction

1- Line 77: Replace “exclude” by “excluded”

PLOS authors have the option to publish the peer review history of their article (what does this mean?). If published, this will include your full peer review and any attached files.

Reviewer #1: No

Reviewer #2: Yes: Kenji Ishida

Reviewer #4: No

Reviewer #5: No

Figure Files:

Data Requirements:

Reproducibility:

References

---

## [Decision Letter · Decision Letter 2]

17 Apr 2023

Dear Dr. Stephane Ranque,

We are pleased to inform you that your manuscript 'Identification of Bulinus forskalii as a potential intermediate host of Schistosoma hæmatobium in Senegal' has been provisionally accepted for publication in PLOS Neglected Tropical Diseases.

Best regards,

Wannaporn Ittiprasert, Ph.D

Academic Editor

Makedonka Mitreva

Section Editor

Reviewer's Responses to Questions

**Key Review Criteria Required for Acceptance?**

**Methods**

-Are the objectives of the study clearly articulated with a clear testable hypothesis stated?

-Is the study design appropriate to address the stated objectives?

-Is the population clearly described and appropriate for the hypothesis being tested?

-Is the sample size sufficient to ensure adequate power to address the hypothesis being tested?

-Were correct statistical analysis used to support conclusions?

-Are there concerns about ethical or regulatory requirements being met?

Reviewer #1: (No Response)

Reviewer #4: (No Response)

Reviewer #5: (No Response)

**Results**

-Does the analysis presented match the analysis plan?

-Are the results clearly and completely presented?

-Are the figures (Tables, Images) of sufficient quality for clarity?

Reviewer #1: (No Response)

Reviewer #4: (No Response)

Reviewer #5: (No Response)

**Conclusions**

-Are the conclusions supported by the data presented?

-Are the limitations of analysis clearly described?

-Do the authors discuss how these data can be helpful to advance our understanding of the topic under study?

-Is public health relevance addressed?

Reviewer #1: (No Response)

Reviewer #4: (No Response)

Reviewer #5: (No Response)

**Editorial and Data Presentation Modifications?**

Reviewer #1: (No Response)

Reviewer #4: (No Response)

Reviewer #5: (No Response)

**Summary and General Comments**

Reviewer #1: My recommendation is the acceptance of the manuscript since all the suggestions and modifications of reviewers were properly addressed by the authors, and the quality of figures, and manuscript text were also improved. I'm adding my final suggestions to the authors so they can add some modifications before publishing the paper.

AUTHOR SUMMARY

Line 52 (manuscript with track changes): change “characterise” to characterize.

INTRODUCTION

Line 81: change “Bulinid” to Bulinus.

Also, check on the manuscript text where to write Bulinus or just B.

METHODS

Line 127: change “The” to the.

Line 129: the coordinates are 15°30W, as is written on the text, or 15°42W as is written on the S1 table?

Line 131: change “analysed” to analyzed.

On the cercarial shedding test, add the Schistosoma morphological identification, if you performed it.

Did you have any documentation approval, from the Senegal government, to do the snail collection? Can you please add it?

DISCUSSION

Line 317: change “economising” to economizing.

Reviewer #4: (No Response)

Reviewer #5: (No Response)

PLOS authors have the option to publish the peer review history of their article (what does this mean?). If published, this will include your full peer review and any attached files.

Reviewer #1: No

Reviewer #4: No

Reviewer #5: No

---

## [Editor Report · Acceptance letter]

4 May 2023

Dear Pr Ranque,

We are delighted to inform you that your manuscript, "Identification of Bulinus forskalii as a potential intermediate host of Schistosoma hæmatobium in Senegal," has been formally accepted for publication in PLOS Neglected Tropical Diseases.

Best regards,

Shaden Kamhawi

co-Editor-in-Chief

Paul Brindley

co-Editor-in-Chief
